# Left-wing support of authoritarian submission to protect against societal threat

**Taylor Winter**[1], **Paul E. Jose**[1], **Benjamin C. Riordan**[2], **Boris Bizumic**[3], **Ted Ruffman**[4], **John A. Hunter**[4], **Todd K. Hartman**[5], **Damian Scarf**[4]*

**1** Department of Psychology, Victoria University of Wellington, Wellington, New Zealand, **2** Centre for Alcohol Policy Research, La Trobe University, Melbourne, Australia, **3** Research School of Psychology, Australian National University, Canberra, Australia, **4** Department of Psychology, University of Otago, Dunedin, New Zealand, **5** Department of Social Statistics, University of Manchester, Manchester, United Kingdom

* damian@psy.otago.ac.nz

## Abstract

New Zealand's Prime Minister, Jacinda Ardern, adopted a "go hard, go early" approach to eliminate COVID-19. Although Ardern and her Labour party are considered left-leaning, the policies implemented during the pandemic (e.g., police roadblocks) have the hallmarks of Right-Wing Authoritarianism (RWA). RWA is characterized by three attitudinal clusters (authoritarian aggression, submission, and conventionalism). The uniqueness of the clusters, and whether they react to environmental change, has been debated. Here, in the context of the pandemic, we investigate the relationship between political orientation and RWA. Specifically, we measured political orientation, support for New Zealand's major political parties, and RWA among 1,430 adult community members. A multivariate Bayesian model demonstrated that, in the middle of a pandemic, both left-leaning and right-leaning individuals endorsed items tapping authoritarian submission. In contrast to authoritarian submission, and demonstrating the multidimensional nature of RWA, we observed the typical relationships between political orientation and authoritarian aggression and conventionalism was observed.

## Introduction

New Zealand's efforts to eliminate the novel coronavirus have been heralded by media outlets around the world [1], and its success has been attributed to Prime Minister Jacinda Ardern's [2] "go hard, go early" approach to containing the virus [3]. In this context, 'going hard' referred to a full national lockdown requiring educational facilities and all but essential businesses to close, while 'going early' meant taking action when New Zealand had recorded just 102 coronavirus cases and 0 deaths [4]. The national lockdown on March 23rd, 2020, temporarily removed the fundamental rights and liberties that New Zealanders typically enjoy, with people instructed to stay at home and only associate with those living in the same household [5]. To enforce these new rules, New Zealand police were given vast new powers such as arresting people who failed to comply with the lockdown rules, setting up roadblocks to question

**Funding:** The author(s) received no specific funding for this work.

**Competing interests:** The authors have declared that no competing interests exist.

drivers about their activities, and the option of calling in the military for additional support. Although the underlying rationale was sound (i.e., to control the spread of COVID-19), with little hyperbole, one could construe the New Zealand government's actions as "authoritarian".

For their part, the public appeared to overwhelmingly support the imposition of these new restrictions in one of the world's most liberal and progressive democracies. For instance, multiple opinion polls reported that 80–90% of the public approved of the government's response to the pandemic [6, 7], which was accompanied by a "soar in popularity" for Prime Minister Jacinda Ardern and her Labour (left-of-center) party [8], as well as a stark increase in general trust of the government, up nearly 25 points (from 59% in March, 2020 to 83% by April, 2020) [9]. The public even went so far as to take an active role to help enforce these new restrictions. For example, the number of people reporting lockdown-rule breakers was so high that the police were forced to establish an online 'COVID-19 breach' report form [10], which then quickly crashed due to the sheer volume of reports being filed [11].

By all accounts, the public willingly responded to the threat of a COVID-19 outbreak by submission to, and enforcement of, the stability, security, and order provided by New Zealand authorities. This behavior–submission to authority figures–is consistent with one of the three defining features of right-wing authoritarianism (RWA). Specifically, scholars have argued that RWA manifests itself as the covariation of *authoritarian submission* (i.e., following the directives of authority figures), *authoritarian aggression* (i.e., punishing those who fail to conform to accepted leaders), and *conventionalism* (i.e., adherence to traditions aimed at preserving the sanctity of the group) [12–16]. In the context of the perceived threat posed by COVID-19, the behavior of the New Zealand public (i.e., almost universal compliance with the strict curbs on fundamental rights and civil liberties) is consistent with authoritarian submission. Moreover, the widespread compliance suggests that those on both the right, and left, of the political spectrum were compliant with the strict lockdown. In the current study, we test this proposition by asking the question: Does the existential threat posed by COVID-19 cause traditionally non authoritarians–that is, those on the left of the political spectrum–to support authoritarian attitudes?

In line with research that focuses on evolutionary and adaptive features of authoritarianism [17, 18], we contribute to the literature by investigating whether those who might traditionally score low on measures of RWA would exhibit signs of this latent construct when faced with a grave, existential threat. Previous studies have documented an overall increase in RWA following destructive natural disasters [19], the 9/11 terrorist attacks [20], and during the COVID-19 pandemic [21]. Given this prior research, and the anecdotal evidence provided above, we hypothesized that the changing relationship between political orientation and RWA, in response to COVID-19, would be specific to authoritarian submission. Although we have witnessed active enforcement of strict lockdown measures, we remain agnostic about whether this will be reflected in support for authoritarian aggression across the political spectrum. Finally, we have no reason to believe that the relationship between political orientation and conventionalism will present with atypical relationships, given that the nature of the COVID-19 threat is not aimed at group homogeneity *per se* but stems from an indiscriminate, yet deadly virus.

In this paper, we thus contribute to the growing literature on the effect of existential threat from COVID-19 on authoritarianism [22, 23]. Rather than examining the negative consequences of authoritarianism on society during times of normative threat such as racial prejudice and intolerance of foreigners [24–27], we instead explore its adaptive and protective implications during times of existential threat. Indeed, Hastings and Shaffer [18] have argued that at our core, humans "are *all* authoritarians", and that the only distinction is each person's specific "sensitivity to threat" (p. 432). While Oesterreich [28] suggests that "even

nonauthoritarians will react in an authoritarian way when a situation overtaxes them emotionally" (p. 283). It may well be that people on different ends of the political spectrum are sensitive to different types of threats–that is, normative threats for high-RWAs and existential threats for high- *and* low-RWAs.

To test our intuitions, we collected data from a sample of 1,431 participants who were living in New Zealand during the country's Alert Level 4 lockdown (March 26th to April 27th, 2020). During this month, participants completed the Duckitt, Bizumic [29] 36-item authoritarianism scale, with subscales that measure submission, aggression, and conventionalism. In addition, participants completed a measure of their political orientation, their support for the country's two major political parties (i.e., the left of center Labour Party and the right of center National Party), and a measure of their fear of coronavirus. To provide an approximation of a pre-coronavirus baseline of the relationship between authoritarianism and political party support, we also utilized the Duckitt, Bizumic [29] scale validation data collected in 2010. We expected that pre-coronavirus data would show negative or null associations between RWA and support for the Labour Party (and positive associations between RWA and affiliation with the National Party); however, during the COVID-19 lockdown, left-leaning individuals would manifest positive associations with authoritarian submission. In line with these predictions, we document a categorically different relationship between authoritarian submission and the other two authoritarian subfactors among left-leaning individuals. Our findings align with Duckitt et al.'s [29] theoretical view that authoritarian submission works to maintain the status quo via social cohesion. That is, the New Zealand public sought to maintain the structure and function of society by following the government's rules that aimed to eliminate the threat of COVID-19.

## Method

### Participants and procedure

We recruited a sample of 1,624 participants, of whom 1,431 (88%) provided data for all questions and were included in our analysis. Participants were living in New Zealand and were recruited through posts on New Zealand news (e.g., The Spinoff, Kiwi Blog, and the NZ Herald) and social networking websites (Facebook) during the country's Alert Level 4 lockdown. In New Zealand, Alert Level 4 was active from March 26th to April 27th, 2020, and consisted of movement restrictions (e.g., stay at home, limit travel to local areas), as well as the closure of all educational facilities and non-essential businesses. To participate, people were asked to click on the study link embedded in advertisements, which took them to a Qualtrics survey with an information sheet and consent form. Once participants had provided informed consent, they completed a 15-minute survey. All procedures were approved by the University of Otago Human Ethics Committee.

The participants who provided complete data had a mean age of 47 (*Standard Deviation (SD)* = 16), and 41% of participants were female. Liberal-conservative political orientation manifested a mean of 3.7 (*SD* = 1.5; range = 1 (liberal) to 7 (conservative)), which fell near the center of the scale, and the Likert scale scores separately for National party and Labour party support were 0.45 (*SD* = 3.2) and -0.43 (*SD* = 3.3), respectively (range = -5 to 5). These descriptive statistics suggest we obtained good representation across the political spectrum, as scale means are close to the center of the political spectrum and variation is large.

Ethics was granted by the University of Otago Human Ethics Committee. Consent was granted in writing, and, due to the sensitive nature of COVID-19 at the time of collection. Participants separately consented to answer COVID-19 related questions.

## Measures

**Demographics.** We collected self-report measures of ethnicity (European NZ, Māori, Pacifica, Asian NZ, Other), age, and gender (male, female, other).

**Right-wing authoritarianism.** The Authoritarianism-Conservatism-Traditionalism Scale (ACT) is a 36-item measure in which participants are asked to rate from 1 (*very strongly disagree*) to 7 (*very strongly agree*) the extent to which they agree with each item [29]. The scale has three subscales: 1) authoritarian aggression (or authoritarianism; i.e., the extent to which one has strong negative attitudes towards people perceived to violate rules and norms; "We should smash all the negative elements that are causing trouble in our society"); 2) conventionalism (or traditionalism; i.e., the extent to which one favours traditional and conventional norms or values; "The 'old-fashioned ways' and 'old-fashioned values' still show the best way to live"); and 3) authoritarian submission (or conservatism; i.e., the extent to which one submits to authorities; "Our country will be great if we show respect for authority and obey our leaders"). All subscales yielded good internal consistency in our sample with Cronbach's alphas >0.80 for authoritarian submission (0.86), authoritarian aggression (0.89), and conventionalism (0.89).

**Political orientation (liberal to conservative).** The political orientation scale was a single item, which asked participants to indicate from 1 (*liberal*) to 7 (*conservative*), how they describe their political beliefs.

**Political support for specific political parties.** The political support scale asked participants to indicate from -5 (*totally oppose*) to +5 (*totally support*) how strongly they support or oppose political parties in the upcoming election, yielding continuous scale measures for all relevant parties. For the purposes of this study, we used support of the two main political parties, who jointly received around 76% of the votes in the 2020 election: Labour (center left and currently leading the NZ government) and National (center right and currently in opposition).

**Fear of COVID-19 scale.** The fear of COVID-19 scale is a seven-item measure of fear of COVID-19 in which participants are asked to rate from 1 (*strongly disagree*) to 7 (*strongly agree*) the extent to which they agree with each item (e.g., "I cannot sleep because I am worried about getting coronavirus-19"). The scale has been validated for use for English-speaking and New Zealand samples [30]. The scale yielded good internal consistency (Cronbach's $\alpha$ = .89).

## Analytic strategy

To prepare the data, we first estimated the content validity of the fear of COVID-19 scale using confirmatory factor analysis (CFA) in *lavaan* [31], written in the R programming language [32]. The fear of COVID-19 scale contains seven questions and therefore can experience problems with collinearity between items, which inflates error when testing a CFA model of a measure composed of more than 3 or 4 items [33]. Consequently, we used item parceling to reduce mis-estimation of redundant error. Thus, the fear of COVID-19 scale was systematically parceled (average of sums of every 4[th] item) into three aggregate variables, two parcels containing two items and a single parcel containing three items [34]. Similarly, the RWA scale was parceled from the original 12 items for each of the three factors to four parcels containing the average of every third item for each subscale. Latent variables for each of the three ACT scales were then produced using the resulting parcels as manifest variables.

After producing latent variables, we used multivariate Bayesian regression using *brms* [35], written in R [32], to 1) test the relationship between political orientation and the RWA subscales, and 2) political affiliation and RWA subscale. Specifically, each of the three RWA factors were treated as separate outcome variables with age, sex, fear of COVID-19, Labour party

support, National party support, and an interaction between fear of COVID-19 and the two-party support variables as predictors. We included fear of COVID as a moderator to control for potential differences between the association of political party support and RWA subscales that could have been influenced by fear (e.g., if National supporters were more prone to elevate their level of submission in the presence of COVD-19 fear). In a follow-up analysis, we used political orientation in lieu of political party support as predictors. This approach was used to affirm that findings are related to political beliefs rather than to idiosyncrasies unique to one or both political parties. We used a weakly informative normal prior with mean 0 and standard deviation of 2 to increase efficiency of sampling. We used the Bayesian approach because it allowed us to make better inferences about the probabilities of our hypothesized effects in the data, which are not generally afforded through a frequentist approach.

## Results

The three factors comprising the RWA scale, and the Fear of COVID-19 scale, all yielded adequate fit with a CFI over 0.90 and an RMSEA less than 0.1 (see S2 Table). Consistent with previous research, political beliefs yielded a 0.54 correlation with National support, and a -0.54 correlation with Labour support, confirming that those participants who are more conservative are more likely to show greater National support and vice versa (Table 1). Similarly, we see that for RWA, aggression, and traditionalism, there is a positive correlation for National support and negative correlation for Labour support. In contrast, submission shows no correlation with National support and a positive correlation with Labour support. Finally, and perhaps of relevance to these otherwise paradoxical findings, COVID-19 fear showed a correlation only with submission and no other authoritarian subscale. COVID-19 fear also showed a positive correlation with Labour support, but a negative correlation with National support.

### Political support and ACT subscales

To test our main hypothesis, we investigated the relationships between political party support and the RWA subscales using multivariate regression (see Table 2). Post-hoc testing using the posterior distributions of National and Labour coefficients was consistent with our hypothesis that supporters of Labour and National would not differ in their ratings on the authoritarian submission subscale (with similar slopes and a less than 95% chance of a difference; see middle panel of Fig 1). That is, those individuals who might typically score low on RWA demonstrated an increase in authoritarian submission in the face of an existential threat. In contrast, for the authoritarian aggression and conventionalism subscales, there was a 99% probability of a difference between

**Table 1. Correlation matrix for key variables used in analyses.**

|  | 1 | 2 | 3 | 4 | 5 | 6 | 7 |
|---|---|---|---|---|---|---|---|
| Political beliefs (1) |  |  |  |  |  |  |  |
| National support (2) | 0.54** |  |  |  |  |  |  |
| Labour support (3) | -0.54** | -0.73** |  |  |  |  |  |
| Submission (4) | 0.17** | -0.01 | 0.22** |  |  |  |  |
| Traditionalism (5) | 0.6** | 0.31** | -0.37** | 0.37** |  |  |  |
| Aggression (6) | 0.57** | 0.45** | -0.43** | 0.42** | 0.61** |  |  |
| RWA (7) | 0.6** | 0.38** | -0.34** | 0.63** | 0.81** | 0.92** |  |
| COVID-19 fear (8) | -0.18** | -0.28** | 0.35** | 0.23** | -0.04 | -0.05 | 0.01 |

n = 1431

** denotes $p < 0.01$

**Table 2. Summary of multivariate regression analysis for RWA subscales and support of political parties.**

| Dependent Variable | Effect | Estimate | Lower | Upper |
|---|---|---|---|---|
| Submission | Age | 0.03 | -0.02 | 0.09 |
| | Gender (male) | -0.16* | -0.28 | -0.05 |
| | Fear of COVID-19 | 0.17* | 0.11 | 0.22 |
| | Labour | 0.39* | 0.31 | 0.46 |
| | National | 0.34* | 0.27 | 0.42 |
| | Labour * Fear | 0.04 | -0.03 | 0.10 |
| | National * Fear | 0.01 | -0.05 | 0.08 |
| Conventionalism | Age | 0.10* | 0.05 | 0.15 |
| | Gender (male) | -0.09 | -0.21 | 0.02 |
| | Fear of COVID-19 | 0.11* | 0.05 | 0.16 |
| | Labour | -0.33* | -0.41 | -0.26 |
| | National | 0.08* | 0.01 | 0.15 |
| | Labour * Fear | 0.06 | -0.01 | 0.13 |
| | National * Fear | 0.06 | 0.00 | 0.12 |
| Aggression | Age | -0.02 | -0.07 | 0.03 |
| | Gender (male) | -0.08 | -0.19 | 0.03 |
| | Fear of COVID-19 | 0.12* | 0.07 | 0.17 |
| | Labour | -0.28* | -0.35 | -0.21 |
| | National | 0.31* | 0.24 | 0.37 |
| | Labour * Fear | 0.07* | 0.01 | 0.13 |
| | National * Fear | 0.03 | -0.03 | 0.09 |

*Note.* N = 1,431.

the slopes (see left and right panels of Fig 1), with National supporters showing markedly higher authoritarian aggression and conventionalism relative to Labour supporters.

Next, we examined the relationship between political party support and the RWA subscales as moderated by fear of COVID-19. As seen in Table 2, for the authoritarian submission sub-scale, we found a non-significant interaction between political support and fear of COVID-19. That is, fear of COVID-19 did not change the strength of the association between level of political support and authoritarian submission (i.e., less than 95% probability of an effect being present). In contrast, for the conventionalism subscale, we found a significant interaction between political support of both Labour and National parties and fear of COVID-19. In this case, fear of COVID-19 amplified the strength of both Labour and National party support in predicting conventionalism (i.e., more than 95% probability of an effect being present). Finally, for the authoritarian submission subscale, we found a significant interaction between political support and fear of COVID for Labour supporters (99% probability of an effect), but not for National party supporters (82% probability of an effect). Thus, Labour party supporters showed a larger effect of COVID-19 fear on authoritarian aggression than did National party supporters. Nevertheless, it should be noted that even Labour supporters with high fear of COVID-19 still maintained lower levels of aggression than National supporters with equivalent fear of COVID-19.

## Political beliefs and RWA subscales

Our analyses regressing RWA subscales on political beliefs assessed whether higher levels of authoritarian submission were predicted by left-wing political orientations during the

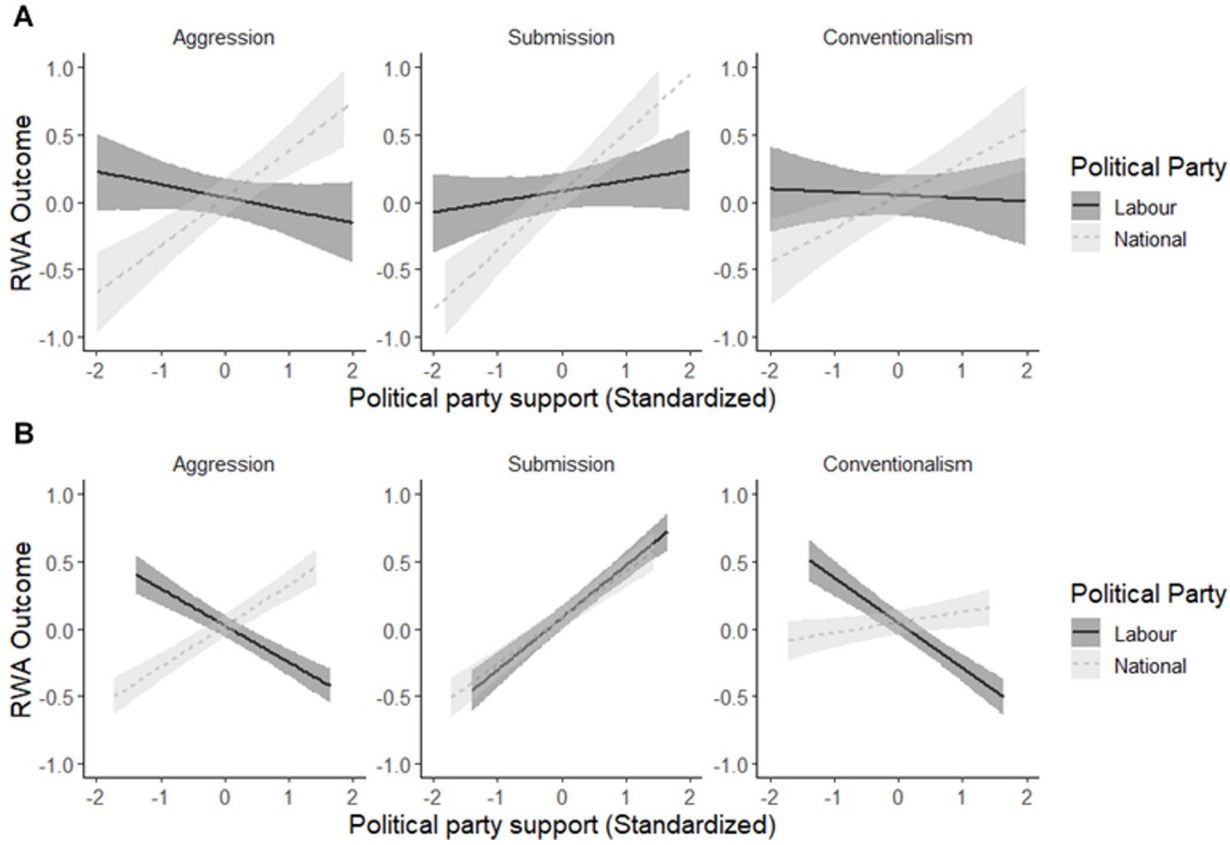

**Fig 1. Level of authoritarianism associated with different levels of support for the Labour and National parties.** Panel A shows results from Duckitt et al.'s [29] data using a similar regression analysis, whereas Panel B shows results from the present data collected in 2020. Error bands represent 95% credible intervals. *Note.* Negative values on the x-axis indicate lack of support for a party, whereas positive numbers indicate support for a party. The y-axis represents the level of response to each subscale identified above each panel where a higher number is a stronger response.

pandemic. As expected, we obtained significant associations between higher levels of conservative political beliefs and all three RWA subscales (Table 3) for the entire sample. Notably, the coefficient for the authoritarian submission component of RWA was much lower, between 0.19 and 0.29, relative to the other two subscales where coefficients ranged from 0.53 to 0.66. This inconsistency supports the prior analysis on political support by suggesting those individuals who are politically liberal reported higher levels of authoritarian submission but not traditionalism or aggression. It is also important to note that all three subscales evidenced positive relationships with fear of the COVID-19 virus.

Both authoritarian submission and aggression had more than a 95% probability of an interaction between COVID-19 fear and political beliefs. Specifically, those individuals reporting more conservative political beliefs also reported higher levels of submission and aggression under conditions of higher levels of COVID-19 fear relative to lower levels of COVID-19 fear. More liberal respondents, by contrast, yielded the same levels of submission and aggression regardless of their levels of fear of COVID-19.

## Discussion

Our findings demonstrate that, within the context of a global pandemic, support for the left-leaning Labour Party was positively associated with support for authoritarian submission. This

**Table 3. Summary of multivariate regression analysis for RWA subscales and liberal-conservative political beliefs.**

| Dependent Variable | Effect | Estimate | Lower | Upper |
|---|---|---|---|---|
| Submission | Age | -0.06 | -0.11 | 0.00 |
| | Gender (male) | -0.35* | -0.46 | -0.25 |
| | Fear of COVID-19 | 0.20* | 0.15 | 0.26 |
| | Conserv political beliefs | 0.27* | 0.22 | 0.32 |
| | Beliefs * Fear | 0.07* | 0.02 | 0.12 |
| Conventionalism | Age | 0.04 | -0.01 | 0.08 |
| | Gender (male) | -0.02 | -0.12 | 0.08 |
| | Fear of COVID-19 | 0.08* | 0.03 | 0.12 |
| | Conserv political beliefs | 0.61* | 0.56 | 0.65 |
| | Beliefs * Fear | 0.01 | -0.03 | 0.05 |
| Aggression | Age | -0.04 | -0.09 | 0.00 |
| | Gender (male) | 0.11* | 0.02 | 0.21 |
| | Fear of COVID-19 | 0.07* | 0.02 | 0.12 |
| | Conserv political beliefs | 0.58* | 0.54 | 0.63 |
| | Beliefs * Fear | 0.07* | 0.03 | 0.12 |

Upper and lower limits denote the 95% credible intervals, that is, we are 95% confident that the true parameter estimate lies within these two values.

* posterior probability < 0.05.

*Note*. N = 1,431.

relationship was comparable to that observed for the supporters of the right-of-center National Party. Importantly, at the same time we observed this relationship for authoritarian submission, we demonstrate the typical relationships between political orientation and endorsement of authoritarian aggression and conventionalism. That is, a positive relationship for National Party support and a negative relationship for Labour Party support. Offering additional support for our findings, we replicated these effects using a non-party-based liberal-conservative political beliefs scale [36]. Finally, both Labour and National supporters yielded the same interaction effects, with COVID-19 fear predicting submission, suggesting that supporters of each political party were similarly motivated by fear of COVID-19.

Our findings are consistent with anecdotal evidence observed in New Zealand, including the spike in popularity for Prime Minister Jacinda Ardern, the widespread usage of the COVID-19 breach report form used to report members of the public in breach of lockdown laws, and the backlash against young people who were, according to the media, not complying with the lockdown rules [37]. Although fronted by Prime Minister Jacinda Ardern, it is likely that participants' conceptualization of an 'authority' when responding to the submission items extended to the scientists and public health experts that fronted much of New Zealand's response, with almost daily news articles written by, or quoting, these scientific authorities. The articles not only supported the need to lockdowns, but also provided evidence to support this position. Thus, submission by Labour Party supporters is consistent with a much wider literature showing that left-leaning individuals tend to believe strongly in scientists and science [38–40].

We assume that Labour Party supporters endorsed the authoritarian submission items because of the potentially dire, existentialist threat, posed by COVID-19. From the lens of the motivational model of authoritarianism, even left-leaning individuals responded to the threat of COVID-19 by supporting adherence to lockdown policies and restrictions because these were perceived as mitigating the imminent threat it posed [15]. An alternative explanation is that support for authoritarian submission by Labour Party supporters is the 'rally around the

flag' effect, with support for those in power increasing during times of stress and turmoil [41, 42]. From this perspective, endorsement of the submission items is simply the result of Labour Party supporters following the lead of Prime Minister Jacinda Ardern. Consistent with the rally around the flag effect, at the same time the current study was conducted, there was a sharp increase in political support for the current prime minister, starting from a 36% prefer- ence in November 2019 and soaring to 62% preference in May 2020 [43]. It is important to note that these explanations are not mutually exclusive, with higher levels of submission likely influenced by both existential threat posed by the pandemic and the rallying of support the government in power.

As outlined by Butler [44], RWA correlates moderately with personality traits and presents somewhat independently of an individual's experienced emotions, which has led to a trait- based versus state-based controversy in the literature. In contrast, Duckitt, Bizumic [29] stipu- late that although there is a personality influence on RWA, RWA is not a strongly stable and unchanging personality dimension [cf. 12, 13, 27]; instead, it is better conceptualized as an latent predisposition that includes three distinct underlying dimensions, which may fluctuate depending on the contextual level of threat to collective security. By identifying an atypical positive relationship between left-leaning political affiliation and authoritarian submission during the COVID-19 pandemic, we provide additional evidence for the motivational model of RWA (i.e., that RWA is malleable and can indeed fluctuate in different environments).

Our findings offer novel insights into RWA research in that we have demonstrated, under certain contexts, authoritarian submission is comparable in people on the left and right of the political spectrum. The approach taken in the current study can be contrasted with the recent work of Costello, Bowes [45]. Rather than utilizing an existing scale, these authors utilized a data-driven approach to identify and develop an authoritarianism scale that is specific to those on the political left. Specifically, they identified the unique tripartite structure of anti-hierarchical aggression (e.g., "The rich should be stripped of their belongings and status"), top-down censor- ship (e.g., "University authorities are right to ban hateful speech from campus"), and anticon- ventionalism (e.g., "People who are truly worried about terrorism should shift their focus to the nutjobs on the far-right"). Although some ideological differences are evident when comparing Costello, Bowes' [45] scale to Duckitt, Bizumic's [29] ACT scale, they both contain subscales that tap submission to authority. Thus, rather than LWA and RWA having either a) completely unique components or, b) being mirror images of one-another, it appears some components are shared by authoritarians on the left and right (i.e., submission), while the remaining components are unique. With regard to COVID-19 mitigation specifically, Manson [46], using a short ver- sion of Costello, Bowes' [45] LWA scale, found that Americans high in either LWA or RWA were more likely, in Spring 2020, to favor many of the same intrusive government mitigation policies, compared to individuals who scored low in both forms of authoritarianism.

## Limitations and future directions

A limitation of our study is that we did not know the original levels of RWA *before* the onset of the COVID-19 pandemic for Labour and National supporters who participated in the current study. We would argue, however, that Duckitt, Bizumic's [29] scale validation data provides an approximation of what the pre-COVID-19 authoritarianism data would have looked like had we been able to collect that baseline data. In addition, with respect to sampling, our snowball methodology does not give a holistic view of the entire population without post-stratification and considered sampling design. Although we have demonstrated important theoretical points of the motivational model, how they present across an entire population may be a further point of study.

Another limitation is that we analyzed data obtained from citizens of a country run by a left-leaning government that has largely mitigated the threat posed by COVID-19. In other words, our phenomenon was confounded by at least two other important contextual factors: 1) the left-of-center orientation of the government in power and, 2) the success of the government in controlling the pandemic. It is unclear whether we would have observed a similar relationship between political orientation and submission had an equally successful right-, rather than left-of-center, government been in power.

And last, Duckitt and Sibley's [15] motivational model stipulates that authoritarian submission is motivated by a desire for social cohesion and society-wide adherence to lockdown restrictions. We did not collect measures to assess motivations for authoritarian submission, and we did not assess perceptions of social cohesion and compliance; thus, we could not test this full sequence of states.

## Conclusions

In conclusion, our findings offer a novel insight into how facets of authoritarianism appear differently to one another in the face of contextual threat, and they suggest that the motivational model of authoritarianism offers a reasonable explanatory mechanism for differences that occur during extreme stress. Specifically, we demonstrated that Labour support was uncharacteristically associated with increased levels of authoritarian submission during the perilous and fear-inducing COVID-19 lock-down period, and we explained this unusual positive association by noting that it occurred among left-leaning citizens in a country run by a left-leaning government. We further substantiated this finding by showing that this effect was not caused by an ideological position unique to Labour supporters, but a phenomenon experienced by politically left-wing/liberal individuals more broadly within New Zealand. Further, fear of COVID-19 did not explain differences between left- and right-leaning individuals in this regard either. In the present climate of COVID-19 in New Zealand, it would seem that submission to the government was perceived to be helpful to the public good, and this belief may have, in fact, functioned as a protective factor allowing a swifter and better coordinated response.

## Supporting information

**S1 Appendix.**
(DOCX)

**S2 Appendix.**
(DOCX)

**S1 Table. Descriptive statistics of demographic variables.**
(DOCX)

**S2 Table. Outcome of confirmatory factor analysis for parcelled ACT scale.**
(DOCX)

## Author Contributions

**Conceptualization:** Taylor Winter, Damian Scarf.

**Investigation:** Taylor Winter, Damian Scarf.

**Methodology:** Taylor Winter, Boris Bizumic.

**Resources:** Damian Scarf.

**Supervision:** Paul E. Jose, Damian Scarf.

**Validation:** Boris Bizumic, Todd K. Hartman.

**Writing – original draft:** Taylor Winter, Damian Scarf.

**Writing – review & editing:** Taylor Winter, Benjamin C. Riordan, Boris Bizumic, Ted Ruffman, John A. Hunter, Todd K. Hartman, Damian Scarf.

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
