## [Decision Letter · Decision Letter 0]

9 Feb 2022

PONE-D-21-21693Left-wing support of authoritarian submission to protect against societal threatPLOS ONE

Dear Dr. Winter,

Thank you for submitting your manuscript to PLOS ONE. After careful consideration, we feel that it has merit but does not fully meet PLOS ONE’s publication criteria as it currently stands. Therefore, we invite you to submit a revised version of the manuscript that addresses the points raised during the review process.

We look forward to receiving your revised manuscript.

Kind regards,

Giray Gozgor, Ph.D.

Academic Editor

PLOS ONE

Journal Requirements:

Additional Editor Comments:

Dear Authors,

Thank you for your submission. Please consider each comment carefully, as reviewers (particularly Reviewer 2) have provided critical issues.

There is a thin line between the R&R and rejection in the light of these reports. Nevertheless, I decided to give you an opportunity to revise your manuscript since I found it interesting.

We are looking forward to receiving your revision in good time.

Best Regards,

Professor Giray Gozgor

Reviewers' comments:

Reviewer's Responses to Questions

**Comments to the Author**

1. Is the manuscript technically sound, and do the data support the conclusions?

Reviewer #1: Partly

Reviewer #2: Partly

2. Has the statistical analysis been performed appropriately and rigorously? 

Reviewer #1: Yes

Reviewer #2: No

3. Have the authors made all data underlying the findings in their manuscript fully available?

Reviewer #1: Yes

Reviewer #2: Yes

4. Is the manuscript presented in an intelligible fashion and written in standard English?

Reviewer #1: Yes

Reviewer #2: Yes

5. Review Comments to the Author

Reviewer #1: This study presents evidence collected during the Spring 2020 lockdown in New Zealand, showing that the authoritarian submission attitudinal cluster of Right-Wing Authoritarianism (RWA) was positively associated with support for both the major conservative party and the major liberal party, whereas the authoritarian aggression and conventionalism clusters of RWA were positively associated with conservatism but negatively associated with liberalism.

Several major issues need to be resolved before this paper is publishable.

The first issue is empirical. Because support for the Labour Party and support for the National Party were negatively associated with each other (correct? I don't see anywhere in this paper a zero-order correlation matrix of all the variables, something that should always be presented in a paper reporting a study of individual differences), yet both were positively related to authoritarian submission, the question arises: what were the political leanings of the respondents who scored *low* on authoritarian submission? Were they middle-of-the-roaders, with opinions in between Labour and National? Were they adherents of parties to the left of Labour? To the right of National? Or were they adherents of parties with platforms orthogonal to the traditional left-right dimension? It appears that the authors can't answer this key question, because they didn't collect enough data. They state that 24% of NZ's voters cast ballots for parties other than Labour or National in the 2020 election. For a study of political psychology, that's a large percentage of your population to leave without the opportunity to express their political preference.

The second issue has to do with the construct of Left-Wing Authoritarianism (LWA), which the authors dismiss with a few sentences in the Discussion. First, although the Manson (2020) study is relevant, the key paper here (already cited 38 times according to Google Scholar), yet which the authors inexplicably fail to cite, is:

Costello, T. H., Bowes, S. M., Stevens, S. T., Waldman, I. D., Tasimi, A., & Lilienfeld, S. O. (2021). Clarifying the structure and nature of left-wing authoritarianism. Journal of Personality and Social Psychology.

This is a solid, thorough, multi-study work of political psychometrics, that (in my view) firmly establishes LWA as a valid construct. The authors of the present study must engage with it seriously, and they must admit that a major limitation of their own study is that they did not measure LWA.

A third issue, which I wouldn't raise if the authors had stuck to description and not ventured into editorializing (lines 290-292), is their repeated, uncritical description of COVID-19 as an "existential threat" that necessitated NZ's lockdown measures. Was a virus with a 0.3% case fatality rate (at its worst) really an existential threat? Sweden imposed only mild restrictions on business and social life, and not only does it still exist, its per-capita COVID death rate wasn't among the highest in Europe. If you're going to editorialize about the lockdowns being a "good thing," then you need to address the complicated cost-benefit analysis entailed by them. The lockdowns caused substantial harms (economic, psychological, and even medical in the form of delayed cancer screenings and so forth) that must be balanced against their benefits in slowing the spread of COVID. But all that would be beyond the scope of this paper. Instead, I suggest deleting the sentence in lines 290-292, and changing the phrase "existential threat" to "perceived existential threat" throughout this paper.

Minor issues of mistaken or missing words:

Line 65, a word is missing between "spectrum" and "the"

Line 271: shouldn't that be "aggression" rather than "submission"?

Line 250: shouldn't that be "aggression" rather than "authoritarianism"?

Reviewer #2: This paper studies the association between partisan ideology and support for authoritarian values in the context of the Covid-19 pandemic in New Zealand. Its core finding is that in the context of the crisis, differences in authoritarianism between individuals subscribing to different ideologies are muted, i.e. left-wing supporters are equally likely to support “authoritarian submission” than right-wing supporters. However, left-wing supporters continue to be less supportive of other dimensions of authoritarianism.

In general, I found the paper to be well written and clearly argued. I also welcome the author’s contribution in terms of presenting and analyzing novel survey data which they collected themselves. The fundamental research question of the paper is interesting and relevant.

However, I see several significant problems with the paper that prevent me from recommending its publication. Some of these issues may be due to different perspectives across disciplines in the social sciences. The authors’ background is more in social psychology, whereas mine is more in political science and sociology. Still, I think there are important issues that need to be discussed – and some of the problems cannot be amended by a review.

The first – and maybe most serious – one relates to the process of data collection. The paper is generally very short on this issue, but it seems that participants to the survey were recruited via online news websites and social media (Facebook), i.e. there is neither quota sampling nor genuine random sampling. The chosen strategy is very likely to create a severely biased sample. While it is perhaps permissible to employ this snow-balling technique of recruitment for pre-tests and – exceptionally – for survey experiments, I am highly doubtful that it can yield reliable estimates for surveys that at least have the aim of approaching representativeness. A thorough description of the distributions of core variables in the survey sample and a comparison with existing population data could have mitigated (or exacerbated?) some of these concerns. In any case, this fundamental flaw in the data collection process would be a “no-go” in high-ranked peer reviewed journals in political science and sociology.

The second major point is the fact that the paper is interested in how the Covid-19 pandemic has (potentially) changed authoritarian attitudes. A significant limitation in that regard (which the authors fully acknowledge) is that they do not have data on these attitudes from before the crisis (or – for obvious reasons – after the crisis). The authors mention a study that could potentially serve as anchoring point, but they do not actually discuss the findings of this study in detail (which seems to be a pre-study anyway). Hence, the central ambition of the paper – to study the effect of the crisis on changing attitudes – can simply not be met. Therefore, the framing of the paper should be adjusted and toned down in that regard. Realistically, it can also study the association between ideology and authoritarian attitudes in the context of the crisis at a specific point in time.

A third issue is about the interpretation of the core finding and the associated theoretical mechanism. The core finding – as I see it – is that the degree of “authoritarian submission” (related to support for Covid-related policy measures) did not differ significantly between supporters from different parties at the height of the first lockdown. The paper interprets this as indicated wide-spread support for authoritarian values in the face of crisis across the board, i.e. the crisis threat overwhelms potential concerns that left-wing people might have against lockdown measures. However, the paper does not take into account a potential “rally behind the flag”-effect that is quite well-known in other domains (e.g. in the case of wars). In crisis moments, citizens tend to rally behind their political leadership, in particular left-wing respondents could become more supportive of whatever “their” left-wing government is doing. The mechanism is then, however, not a changing ideology, but party loyalty.

Besides these major points, I have a few smaller issues:

- The data and the regression analysis work with a rather reduced set of socio-economic control variables. What is in particular missing is information about respondents’ educational background, their socio-economic position, their income, and potentially their labor market position. These are quite standard things to control for in studies of attitudes and preferences in political science, so I was surprised to not find them here.

- It would be good to have the detailed wordings of all the items in both the scales on authoritarianism and fear of Covid, at least in the appendix.

- The paper would benefit from putting the case of New Zealand into a more comparative context. What kind of broader implications can researchers draw from studying this relatively peculiar and special case?

6. PLOS authors have the option to publish the peer review history of their article (what does this mean?). If published, this will include your full peer review and any attached files.

Reviewer #1: **Yes: **Joseph H. Manson

Reviewer #2: No

---

## [Author Response · Author response to Decision Letter 0]

29 Mar 2022

We thank the editor and reviewers for their time in assessing this manuscript. We have attached a document in our resubmission that goes through each major point raised by reviewers and contains our direct response on how we sought to address the recommendation in the manuscript. We now feel all points raised by reviewers have been appropriately addressed in the new manuscript, resulting in a substantially more robust study.

---

## [Decision Letter · Decision Letter 1]

2 May 2022

PONE-D-21-21693R1Left-wing support of authoritarian submission to protect against societal threatPLOS ONE

Dear Dr. Winter,

Thank you for submitting your manuscript to PLOS ONE. After careful consideration, we feel that it has merit but does not fully meet PLOS ONE’s publication criteria as it currently stands. Therefore, we invite you to submit a revised version of the manuscript that addresses the points raised during the review process.

We look forward to receiving your revised manuscript.

Kind regards,

Giray Gozgor, Ph.D.

Academic Editor

PLOS ONE

Journal Requirements:

Additional Editor Comments:

Dear Authors,

Thank you for your resubmission. Reviewers have provided minor comments to revise your manuscript.

Please also consider other papers to focus on the economic effects on populism, such as

Gozgor, G. (2022). The role of economic uncertainty in the rise of EU populism. Public choice, 190(1), 229-246.

Guriev, S. (2018). Economic drivers of populism. American Economic Review Papers and Proceedings, 108(5), 200–203.

Margalit, Y. (2019). Economic insecurity and the causes of populism, reconsidered. Journal of Economic Perspectives, 33(4), 152–170.

Rodrik, D. (2018). Is populism necessarily bad economics? American Economic Review Papers and Proceedings, 108(5), 196–199.

Rodrik, D. (2021). Why does globalization fuel populism? Economics, culture, and the rise of right-wing populism. Annual Review of Economics, 13, 133–170.

Reviewers' comments:

Reviewer's Responses to Questions

**Comments to the Author**

1. If the authors have adequately addressed your comments raised in a previous round of review and you feel that this manuscript is now acceptable for publication, you may indicate that here to bypass the “Comments to the Author” section, enter your conflict of interest statement in the “Confidential to Editor” section, and submit your "Accept" recommendation.

Reviewer #1: (No Response)

2. Is the manuscript technically sound, and do the data support the conclusions?

Reviewer #1: Yes

3. Has the statistical analysis been performed appropriately and rigorously? 

Reviewer #1: Yes

4. Have the authors made all data underlying the findings in their manuscript fully available?

Reviewer #1: Yes

5. Is the manuscript presented in an intelligible fashion and written in standard English?

Reviewer #1: Yes

6. Review Comments to the Author

Reviewer #1: This revision is almost ready for publication. I suggest restoring the reference to the Manson (2020) paper, by adding the following sentence, in line 322, after the sentence that ends "...while the remaining components are unique":

"With regard to COVID-19 mitigation specifically, Manson (##), using a short version of Costello, Bowles' (##) LWA scale, found that Americans high in either LWA or RWA were more likely, in Spring 2020, to favor many of the same intrusive government mitigation policies, compared to individuals who scored low in both forms of authoritarianism."

7. PLOS authors have the option to publish the peer review history of their article (what does this mean?). If published, this will include your full peer review and any attached files.

Reviewer #1: No

---

## [Author Response · Author response to Decision Letter 1]

3 May 2022

We have addressed all minor revisions raised by the editor and reviewer.

The exact changes are captured in the attached documentation.

---

## [Decision Letter · Decision Letter 2]

1 Jun 2022

Left-wing support of authoritarian submission to protect against societal threat

PONE-D-21-21693R2

Dear Dr. Winter,

We’re pleased to inform you that your manuscript has been judged scientifically suitable for publication and will be formally accepted for publication once it meets all outstanding technical requirements.

Kind regards,

Lorien Shana Jasny

Academic Editor

PLOS ONE

Additional Editor Comments (optional):

I think this will make a great contribution to the literature. One thing to note in proofs - in your abstract and introduction, I think you want to say 'in contrast' rather than 'in contract.'

Reviewers' comments:

Reviewer's Responses to Questions

**Comments to the Author**

1. If the authors have adequately addressed your comments raised in a previous round of review and you feel that this manuscript is now acceptable for publication, you may indicate that here to bypass the “Comments to the Author” section, enter your conflict of interest statement in the “Confidential to Editor” section, and submit your "Accept" recommendation.

Reviewer #1: All comments have been addressed

2. Is the manuscript technically sound, and do the data support the conclusions?

Reviewer #1: Yes

3. Has the statistical analysis been performed appropriately and rigorously? 

Reviewer #1: Yes

4. Have the authors made all data underlying the findings in their manuscript fully available?

Reviewer #1: Yes

5. Is the manuscript presented in an intelligible fashion and written in standard English?

Reviewer #1: Yes

6. Review Comments to the Author

Reviewer #1: (No Response)

7. PLOS authors have the option to publish the peer review history of their article (what does this mean?). If published, this will include your full peer review and any attached files.

Reviewer #1: No

---

## [Editor Report · Acceptance letter]

21 Jun 2022

PONE-D-21-21693R2 

Left-wing support of authoritarian submission to protect against societal threat 

Dear Dr. Winter:

I'm pleased to inform you that your manuscript has been deemed suitable for publication in PLOS ONE. Congratulations! Your manuscript is now with our production department. 

Kind regards, 

on behalf of

Dr. Lorien Shana Jasny 

Academic Editor

PLOS ONE